# Compressed computations using wavelets for hidden Markov models with continuous observations

**Luca Bello[1], John Wiedenhöft [2], Alexander Schliep [1,3] ***

**1** Computer Science and Engineering, University of Gothenburg, Chalmers, Gothenburg, Sweden,
**2** Scientific Core Facility Medical Biometry and Statistical Bioinformatics, University Medical Center
Göttingen, Göttingen, Germany, **3** Faculty of Health Sciences, B-TU Cottbus-Senftenberg, Cottbus, Germany

☯ These authors contributed equally to this work.
* alexander.schliep@cse.gu.se, alexander@schlieplab.org

Compressed computations using wavelets for
hidden Markov models with continuous
observations. PLoS ONE 18(6): e0286074. https://
doi.org/10.1371/journal.pone.0286074

**Data Availability Statement:** All data used in the
manuscript is simulated. Software to generate the
data is distributed along with the open-source
implementation of the method at https://github.
com/lucabello/wavelet-hmms.

## Abstract

Compression as an accelerant of computation is increasingly recognized as an important
component in engineering fast real-world machine learning methods for big data; c.f., its
impact on genome-scale approximate string matching. Previous work showed that com-
pression can accelerate algorithms for Hidden Markov Models (HMM) with discrete observa-
tions, both for the classical frequentist HMM algorithms—Forward Filtering, Backward
Smoothing and Viterbi—and Gibbs sampling for Bayesian HMM. For Bayesian HMM with
continuous-valued observations, compression was shown to greatly accelerate computa-
tions for specific types of data. For instance, data from large-scale experiments interrogating
structural genetic variation can be assumed to be piece-wise constant with noise, or, equiva-
lently, data generated by HMM with dominant self-transition probabilities. Here we extend
the compressive computation approach to the classical frequentist HMM algorithms on con-
tinuous-valued observations, providing the first compressive approach for this problem. In a
large-scale simulation study, we demonstrate empirically that in many settings compressed
HMM algorithms very clearly outperform the classical algorithms with no, or only an insignifi-
cant effect, on the computed probabilities and infered state paths of maximal likelihood. This
provides an efficient approach to big data computations with HMM. An open-source imple-
mentation of the method is available from https://github.com/lucabello/wavelet-hmms.

## Introduction

Hidden Markov Models (HMM) [1] are still one of the most widely used machine learning
methods across an extensive range of data modalities—texts to images, time-series of hand
movements to weather events—and virtually all disciplines analyzing data, from political sci-
ence [2] to education [3] and computer security [4] to autonomous vehicles [5].

Their simple structure, i.i.d. observations generated in each state of the underlying Markov
chain and Markov dynamics between states, is on the one hand powerful enough for black-
box modeling and on the other hand amenable to handcrafting models [6] reflecting expert

**Funding:** The author(s) received no specific funding for this work.

**Competing interests:** The authors have declared that no competing interests exist.

knowledge from the application domain. Their immense success in bioinformatics is largely owed to the latter.

Another aspect driving their popularity is computational efficiency. The standard algorithms—Rabiner's three problems [1]—for computing likelihood under a model, finding the optimal state sequence given an observation with the Viterbi algorithm, and each iteration of the maximum likelihood (ML) estimation of parameters, i.e., the iterations of the Baum-Welch training, are linear in the length of the observation sequence $T$, more exactly $O(N^2 \cdot T)$ where $N$ is the number of states. A tighter bound can be found if the model is sparse, or more specifically, when all out-degrees of the states are bounded by a small constant as for example in profile HMMs [7].

Still, with sufficient amounts of data even a linear algorithms might not deliver running times sufficiently fast in practice. This is certainly the case when analyzing biological sequences, where for example popular gene finding models have hundreds of states. The insight that the combination of a small alphabet—the four characters `A`, `C`, `G`, and `T` representing the four constituent nucleotides of DNA molecules—and large sequence length $T$, up to 248 million for the longest Human chromosome, forces a high degree of repetitiveness in the sequences [8] suggested that exploiting this repetitiveness in computations may achieve a large impact.

Mozes et al. [9] recognized that for discrete observations rewriting HMM-computations as matrix multiplications of transition-emission operators $M^\sigma$, similar to [10] and observable operator models [11], allowed them to pre-compute partial products representing a frequently occurring sequence pattern and thus, with a sufficiently compressible sequence, achieve considerable speed-ups for likelihood, Viterbi and Baum-Welch training.

Also for discrete observations, Mahmud et al. [12] used transition-emission operators to substantially accelerate Forward-Backward Gibbs (FBG) sampling [13], a rapidly converging sampler for computations with fully Bayesian HMM, the *fourth* problem for HMM. There, marginal state probabilities conditioned on the data provides a robust alternative to locally optimal ML estimation followed by Viterbi-computation.

Clearly, there is no trivial path to extend the work from discrete to continuous-valued observations, but a specific type of important data naturally suggested a workable approach. A popular type of experiment in genetics identifies chromosomal duplications and deletions, changes where segments from several hundreds to many thousands of nucleotides of length are lost or added. The data coming from array CGH [14], SNP array platforms [15] or high-throughput sequencing [16] indicates the abundance of DNA vs. genomic location. In absence of any errors the data would be integer-valued and piecewise constant; the observed data is a noisy version of that ideal. The discrete underlying states and the large number of observations between changes explains why HMMs are a very suitable model for this data [17] and suggests a compressive computation approach similar in spirit to run-length encoding in the discrete case. The fundamental assumption is that a state change from $t$ to $t + 1$ is prohibited when consecutive observations $x_t$ and $x_{t+1}$ are close. This assumption was the basis for a greedy clustering algorithm and adapted sampling procedures leading to a substantial improvement [18] in the running times of the FBG sampler for continuous-valued observations with negligble errors in the estimation. Later, Wiedenhoeft et al. [19] proposed a principled approach based on wavelets, in which the breakpoints, i.e. the position of discontinuities in the wavelet reconstruction, defined the boundaries of segments or blocks; see section on wavelets for details. Highly optimized algorithms and data structures for computing such breakpoints [20] helped to achieve several-thousand fold acceleration and greatly improved convergence of the sampler allowing Bayesian computation even for genome-scale data at single-nucleotide resolution [21], thereby settling compressive computation for the fourth problem.

### Novelty and outline

In the following, we introduce the first compressive approach for Rabiner's three original problems [1]—likelihood computation, computing Viterbi paths, and Maximum-likelihood estimation with the Baum-Welch algorithm—for Hidden-Markov Models with continous emissions based on compression obtained with wavelets. We review the basics of the wavelet compression and present adapted equations for HMM computations in this compressive setting. With a large-scale evaluation study, we demonstrate the large improvements in running times for data which is amenable to this compressive approach, namely piece-wise constant with noise, or, equivalently, data generated by HMM with separable means and high self-transition probabilities. An open-source implementation of the method is available from https://github.com/lucabello/wavelet-hmms/tree/v1.0-thesis.

## Materials and methods

First, let us fix some notation for introducing the models and (un-)compressed computations. Random variables will be denoted as upright letters ($x$ or $X$), their instantiated values in italics ($x$ or $X$), and their domains in script ($\mathcal{X}$). To avoid notational clutter, we denote density and mass functions as

$$p(x \mid y) := f_{X \mid Y=y}(x),$$

where the names of the random variables involved can be inferred from their instantiations. Sometimes, instead of evaluating this function at $x$, we have a specific value $j$ from context, which would make inferring the associated random variable impossible. We therefore use the notation

$$p(x = j \mid y) := f_{X \mid Y=y}(j),$$

for disambiguation, where $j \in \mathcal{X}$. Accordingly, fixed values are denoted like $p(x = 1 \mid y)$ and $p(x \mid y = 5)$; note that the first expression explicitly denotes a density and the second a likelihood function. Indeed, being algebraically equivalent, we also have

$$p(x \mid y) := \mathcal{L}(y \mid X = x).$$

Importantly, note that Bayes' theorem holds not only for probability measures, but for densities and mass functions as well:

$$f_{X \mid Y=y}(x) = \frac{f_{Y \mid X=x}(y)}{f_Y(y)} f_X(x) \quad \Rightarrow \quad p(x \mid y) = \frac{p(y \mid x)}{p(y)} p(x).$$

In the following we introduce the three fundamental algorithms we will later accelerate with the wavelet-based compression.

### The three classical HMM problems

It's important to define some notation to properly describe the problems ahead:

- $q_t \in \mathcal{Q}$ denotes the hidden state at position $t$, and $q_s^t = (q_s, \ldots, q_t) \in \mathcal{Q}^{t-s+1}$ denotes a *state sequence*, also called (or *state path*, *generating path*). For the length $T$ of the data, we define $q = q_1^T$ for convenience.

- $N = |\mathcal{Q}|$, the number of states of the HMM;

- $A = \{A_{ij}\} \in \mathcal{A}$, $1 \leq i, j \leq N$, the transition matrix of the Markov process on $\mathcal{Q}$.

- $y_t \in \mathcal{Y}$ denotes the observed value at position $t$. Analog to the definition of $q_s^t$, $y_s^t$ is a partial observation sequence $y_s^t = (y_s, \ldots, y_t) \in \mathcal{Y}^{t-s+1}$ and $y = y_1^T$.

- $\theta_j$ denotes the parametrization of the emission distribution of $y_t$ in state $j$. Due to their importance, we denote the likelihood functions of emissions as $L_j(t) := p(y_t | q_t = j, \theta_j)$.

- $\pi = \{\pi_i\}$, the initial state distribution.

The model will be often indicated through the compact notation

$$\lambda = (N, A, \theta, \pi). \tag{1}$$

**Evaluation problem.** The evaluation problem concerns measuring how well a specific sequence of observations is represented by a given model, through the computation of the probability that the observed sequence was produced by the model. Given an observation sequence $y$ and a model $\lambda$, the goal is to compute its likelihood $p(y|q, \lambda)$.

The standard algorithm used to solve the evaluation problem is the *forward* algorithm. The key element is the forward variable $\alpha_t(i)$, defined as the joint probability of observing the sequence up to time $t$ and being in state $i$ at time $t$

$$\alpha_t(i) = p(y_1^t, q_t = i \mid \lambda). \tag{2}$$

Through induction, the following procedure can be defined:

$$\alpha_1(i) = \pi_i L_i(1), \qquad 1 \leq i \leq N \tag{3a}$$

$$\alpha_{t+1}(j) = \left[ \sum_{i=1}^{N} \alpha_t(i) A_{ij} \right] L_j(t+1), \qquad 1 \leq t \leq T - 1, \quad 1 \leq j \leq N \tag{3b}$$

$$p(y \mid \lambda) = \sum_{i=1}^{N} \alpha_T(i). \tag{3c}$$

Looking at the computational complexity, the number of calculations for each observation is $N^2$ ($N$ per each state); repeating this for the whole sequence length gives a complexity that is $\mathcal{O}(T \cdot N^2)$.

The backward-variables $\beta_t(j) = := p(y_{t+1}^T \mid q_t = j)$ give the probability of observing the remainder of the observation sequence starting from state $j$ at time $t$ and lend themselves to computation with a similar dynamic programming scheme as for the $\alpha_t(i)$.

**Decoding problem.** The decoding problem deals with the computation of the most likely generating path of an observation sequence for a given model; formally, this means finding the maximization of $p(q|y, \lambda)$. The standard approach is to use the *Viterbi* algorithm: it follows a similar strategy to the forward algorithm, but applying a maximization in place of the summation,

$$\delta_t(i) = \max_{q_1^{t-1}} p(q_1^{t-1}, q_t = i, y_1^t \mid \lambda). \tag{4}$$

The most likely path will be the argument of this maximization over all the states considering the whole observations sequence; it can be defined in the notation as $\psi$. Through induction

it is possible to write the following equations:

$$\delta_1(i) = \pi_i L_i(1), \qquad 1 \leq i \leq N$$
$$\psi_1 = 0$$

(5a)

$$\delta_t(j) = \max_{1 \leq i \leq N}[\delta_{t-1}(i)\ A_{ij}]L_j(t), \qquad 2 \leq t \leq T, \quad 1 \leq j \leq N$$
$$\psi_t(j) = \arg\max_{1 \leq i \leq N}[\delta_{t-1}(i)\ A_{ij}], \qquad 2 \leq t \leq T, \quad 1 \leq j \leq N$$

(5b)

$$P^* = \max_{1 \leq i \leq N}[\delta_T(i)]$$
$$q_T^* = \arg\max_{1 \leq i \leq N}[\delta_T(i)]$$

(5c)

$$q_t^* = \psi_{t+1}(q_{t+1}^*), \qquad t = T-1, T-2, \cdots, 1.$$

(5d)

**Training problem.** The goal is to find the model $\lambda$ that maximizes $p(y|\lambda)$; a popular technique is the *Baum-Welch* method, which starts from initial parameter estimates and iteratively performs reestimations of the parameters to improve the likelihood.

This algorithm introduces a new key variable: $\xi(i, j)$, the probability of being in state $i$ at time $t$ and in state $j$ at time $t + 1$

$$\xi_t(i, j) = p(q_t = i, q_{t+1} = j \mid y, \lambda).$$

(6)

It can be useful to express this equation using the forward and backward variables. In fact, the forward variable $\alpha_t(i)$ accounts for the observations from the first one up to $y_t$ in state $i$; the backward variable $\beta_{t+1}(j) := p(y_{t+1}^T \mid q_t = j)$ does the complementary job, considering the observation sequence starting in state $j$ and from observation $y_{t+1}$ up to the last one. The step between $t$ and $t + 1$ has been left out: to tie the two variables, it is necessary to include the probability of transitioning from state $i$ to $j$ and observing $y_{t+1}$, which is $A_{ij}L_j(t + 1)$. The new formulation of $\xi_t$ can be written as

$$\xi_t(i, j) = \frac{\alpha_t(i)A_{ij}L_j(t+1)\beta_{t+1}(j)}{p(y \mid \lambda)} = \frac{\alpha_t(i)A_{ij}L_j(t+1)\beta_{t+1}(j)}{\sum_{p=1}^N \sum_{q=1}^N \alpha_t(p)A_{pq}L_q(t+1)\beta_{t+1}(q)}.$$

(7)

We define the $\gamma_t(i)$ variables as the probability of being in state $i$ at time $t$ while emitting the observation sequence. This can be expressed in terms of the $\xi_t$ as

$$\gamma_t(i) = \sum_{j=1}^N \xi_t(i, j).$$

(8)

Note that the sum $\gamma_t(i)$ over $t$ can be interpreted as the expected number of times that the state $i$ is visited, or equivalently as the expected number of transitions out of state $i$ (if we exclude the last observation at time $T$),

$$\sum_{t=1}^{T-1} \gamma_t(i) = \text{expected number of transitions out of state } i.$$

(9)

Similarly, the sum of $\xi_t(i,j)$ over $t$ can be interpreted as the expected number of transitions from $i$ to $j$

$$\sum_{t=1}^{T-1} \xi_t(i,j) = \text{expected number of transitions from state } i \text{ to } j. \tag{10}$$

These interpretations lead to the definition of two reestimation formulas for the initial distribution and the transition probabilities

$$\bar{\pi}_i = \gamma_1(i), \text{and} \tag{11a}$$

$$\bar{a}_{ij} = \frac{\sum_{t=1}^{T-1} \xi_t(i,j)}{\sum_{t=1}^{T-1} \gamma_t(i)}. \tag{11b}$$

The HMMs that have been considered throughout this thesis work have continuous emission densities. While the model formalism allows finite mixtures of log-concave or elliptically symmetric multi-variate densities, we restricted ourselves to a single univariate Gaussian as emission densities, i.e., in state $j$ we have

$$L_j(y) = \mathfrak{N}(y|\mu_j, \sigma^2), \tag{12}$$

where $\mu_j$ is the mean and $\sigma^2$ is the variance of the Gaussian distribution associated with the state $j$. Thus, the reestimation formulas are given as:

$$\bar{\mu}_j = \frac{\sum_{t=1}^{T} \gamma_t(j) \cdot y_t}{\sum_{t=1}^{T} \gamma_t(j)} \tag{13a}$$

$$\bar{\sigma^2}_j = \frac{\sum_{t=1}^{T} \gamma_t(j) \cdot (y_t - \mu_j)^2}{\sum_{t=1}^{T} \gamma_t(j)} \tag{13b}$$

Applying the reestimation formulas (11a), (11b), (13a) and (13b) produces a reestimated model $\bar{\lambda}$; the Baum-Welch algorithm guarantees that either the original model $\lambda$ is a critical point of the likelihood function (the result would be $\lambda = \bar{\lambda}$) or the model $\bar{\lambda}$ is more likely than the previous one, meaning that $p(y \mid \bar{\lambda}) > p(y \mid \lambda)$. The iteration of this procedure converges to a local maximum and produces a maximum likelihood estimate of the model, providing a solution to the training problem.

## Wavelet compression

In the following we briefly introduce wavelet shrinkage and transform and the resulting block structure enabling compressed HMM computations. Haar wavelet shrinkage is a powerful regression method for univariate time series $y = f + \epsilon$, where $f$ is a piecewise constant function and $\epsilon$ is a vector of centered homoscedastic Gaussian noise of variance $\sigma^2$. If the latter is known, wavelet coefficients whose absolute value is below the *universal threshold* $\lambda_u = \sqrt{2\ln T}\sigma$ can be attributed to noise and removed, yielding a minimax estimator $y = \hat{f} + \hat{\epsilon}$ [22]. The coefficients of the Haar wavelet transform can be computed in-place and in linear time [23, 24], and the variance can conveniently be estimated from the finest detail wavelet coefficients themselves.

Data generated by a homoscedastic Gaussian HMM can be treated in a similar fashion, with discontinuities in $\hat{f}$ marking the approximate location of state transitions; for details, see

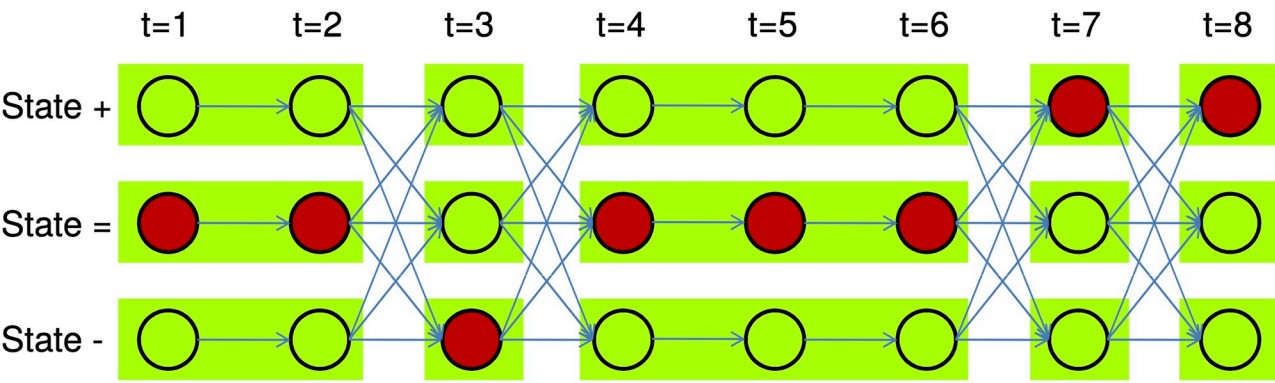

**Fig 1. An example of a compressed trellis for a three-state model with states labeled +, − and =.** Within each block, only self-transitions are allowed. Relevant variables such as the forward or backward variables are computed for each block instead of for each individual observation. Our evaluation demonstrates that for compressible input, i.e., input with large average block sizes, the compressed computation has a neglible impact on accuracy while accelerating computations greatly, in particular for models with many states.

[25]. Note that despite the fact that we can expect $\hat{f}[t] \approx \mu_{q[t]}$, $\hat{f}$ cannot reliably be used to identify $q$ directly. However, since the normal distribution is in the exponential family, sufficient statistics between the discontinuities in $\hat{f}$ can be used in likelihood computations required for inference of HMM parameters, assuming the generating Markov process does not switch states in between these positions, or that at least the contribution of such transitions is negligible (weak path assumption [18]). Blocks are defined as contiguous subsequences of observations respectively hidden states where no state switches are allowed; cf. Fig 1.

While such blocks of sufficient statistics can be precomputed for homoscedastic HMM, in heteroscedastic settings wavelet shrinkage does not yield a minimax estimator, but can still be used for compression, in that the smallest emission variance can be used for thresholding, at the cost of undercompression in high-variance regions. Unfortunately, estimating this in a manner similar to wavelet regression is challenging, since it is unknown which finest detail coefficients come from low-variance segments. In order to estimate HMM parameters, a data structure is required to quickly yield blocks of sufficient statistics at arbitrary variance thresholds, i.e. different compression levels, without re-estimating $\hat{f}$ for every given $\sigma^2$. For this purpose, we have previously developed a highly efficient data structure called a *breakpoint array* with a linear time constructor [20, 25] based on the lifting scheme [23, 24].

## Compressed computation for HMM

To perform the compressed computations, the classic equations have to be adapted to the compression scheme introduced in the previous section; see Fig 1 for a schematic view. Restructuring the forward algorithm does not alter the interpretation of the forward variables. For a block of observations, $\alpha_w(i)$ is the approximation of the uncompressed forward variable at the end of the $w$-th block, and $E_w(j)$ the exponent resulting from considering the length of the block and summary statistics, yielding

$$\alpha_1(i) = \pi_i e^{E_1(i)} \text{ for } 1 \leq i \leq N, \tag{14a}$$

$$\alpha_w(j) = \left[\sum_{i=1}^{N} \alpha_{w-1}(i) A_{ij}\right] e^{E_w(j)} \text{ for } 1 \leq w \leq W, 1 \leq j \leq N, \text{ and} \tag{14b}$$

$$p(y \mid \lambda) \approx \sum_{i=1}^{N} \alpha_W(i). \tag{14c}$$

Here, the term $e^{E_w(j)}$ above results from making the Gaussian assumption explicit and rewriting the regular induction for forward variables adapted to $(n-1)$ self-transitions and one non-self transitions in a block,

$$\alpha_w(j) = \sum_{i=1}^{N} \left[ \alpha_{w-1}(i)a_{ij} \right] a_{jj}^{n_w-1} \prod_{k=1}^{n_w} b_j(y_{w,k}). \tag{15}$$

and collecting as many terms as possible in the exponential, which additionally improves numerical stability, leading to

$$E_w(j) := \frac{2\mu_j \Sigma_{1,w} - \Sigma_{2,w}}{2\sigma_j^2} + K(n_w, j), \quad \text{and} \tag{16a}$$

$$K(n_w, j) := (n_w - 1)\log(a_{jj}) - n_w \left( \log(\sigma_j) + \frac{\mu_j^2}{2\sigma_j^2} + \frac{1}{2}\log(2\pi) \right). \tag{16b}$$

Both the *backward* and *Viterbi* algorithms require similar transformations, using block summary information in place of the individual observations.

The *Baum-Welch* algorithm is more complex than the others, requiring more variables in the parameter reestimation. It is important to remember that both the compressed forward and backward variables refer to the end of a block. Consequently, different situations result based on computing $\xi$ inside or outside a block. For the purpose of rewriting the reestimation equations, it is useful to define the $\xi$ variable for a block in the following way

$$\xi_w(i,j) = \sum_{t \in Y_w} \xi_t(i,j) = \tag{17}$$

$$= \frac{1}{p(y \mid \lambda)} \cdot \begin{cases} (n_w - 1)\alpha_w(i)\beta_w(i) + \alpha_w(i)A_{ij}e^{E_{w+1}(j)}\beta_{w+1}(j) \ , & \text{for } i = j \wedge w \neq W \\ (n_W - 1)\alpha_W(i)\beta_W(i) \ , & \text{for } i = j \wedge w = W \\ \alpha_w(i)A_{ij}e^{E_{w+1}(j)}\beta_{w+1}(j) \ , & \text{for } i \neq j \wedge w \neq W \\ 0 \ , & \text{for } i \neq j \wedge w = W. \end{cases} \tag{18}$$

Moving forward, it is interesting to note that by interpreting $\gamma_t(i)$ as the probability of visiting the state $i$ at time $t$, the variable is also constant over $t$ inside a block (also implied from the result above); this means that for any $t$ inside a block, any $\gamma_t(i)$ can be representative for the whole block, say

$$\gamma_t(i) = \frac{\alpha_t(i)\beta_t(i)}{p(y \mid \lambda)} = \frac{\alpha_w(i)\beta_w(i)}{p(y \mid \lambda)}. \tag{19}$$

It is worth noting that this reformulation correctly maintains the definition given in (8). For convenience it is useful to define $\gamma_w(i)$ as the representative value for a block, which means that for all the $t$ associated with a block, $\gamma_w(i) = \gamma_t(i)$.

The reestimation formulas Eqs (11a), (11b), (13a) and (13b) can be updated with the new definitions of $\xi_t(i, j)$ and $\gamma_t(i)$, yielding

$$\bar{\pi}_i = \gamma_1(i), \text{ and} \tag{20a}$$

$$\bar{a}_{ij} = \frac{\sum_{w=1}^{W} \xi_w(i,j)}{\sum_{w=1}^{W} [n_w \gamma_w(i)] - \gamma_W(i)}. \tag{20b}$$

The mean and standard deviation reestimations follow a slightly more complex reformulation, both for the general mixture and the single Gaussian distribution. In particular, Eq (13a) multiplies the single observation value by the respective $\gamma_t(i)$. Since $\gamma_t(i)$ is constant inside a block, the equation can be rewritten as

$$\bar{\mu}_j = \frac{\sum_{t=1}^{T} \gamma_t(j) y_t}{\sum_{t=1}^{T} \gamma_t(j)} = \frac{\sum_{w=1}^{W} \gamma_w(j) \sum_{k=1}^{n_w} y_{w,k}}{\sum_{w=1}^{W} \gamma_w(j) \cdot n_w} = \frac{\sum_{w=1}^{W} \gamma_w(j) \cdot \Sigma_{1,w}}{\sum_{w=1}^{W} \gamma_w(j) \cdot n_w}. \tag{21}$$

The same reasoning applies to the variance reestimation, yielding

$$\bar{\sigma}_j^2 = \frac{\sum_{t=1}^{T} \gamma_t(j)(y_t - \mu_j)^2}{\sum_{t=1}^{T} \gamma_t(j)} = \frac{\sum_{w=1}^{W} \gamma_w(j)[\Sigma_{2,w} - 2\bar{\mu}_j \Sigma_{1,w} + n_w \bar{\mu}_j^2]}{\sum_{w=1}^{W} \gamma_w(j) \cdot n_w}. \tag{22}$$

## Implementation

Our implementation named WaHMM (**W**avelets **HMM**) adapts the classic algorithms to the compressed structures: instead of operating on every individual observation in a sequence, it takes advantage of the blocks' summary information. The wavelet compression is performed using parts of the software HaMMLET https://schlieplab.org/Software/HaMMLET/.

WaHMM is written in C++ for better efficiency and a smaller memory footprint. To offer a simpler interface to the user, some Python scripts are provided for model creations, usage of the actual tool, and for analysis and graphical display of results. The software implements HMM with univariate Gaussian state densities. The Python framework Pomegranate (https://github.com/jmschrei/pomegranate) is used to generate data from the model, through the script generate_data.py.

Note that in WaHMM probabilities are represented in a logarithmic space; this not only scales the [0, 1] interval to $(-\infty, 0]$, easing the burden of numerical precision, but also transforms all the products into summations, which are much easier and faster to perform.

## Results

In the following we introduce the simulation study which varies compressibility and thus indicates speed-ups of our method for various scenarios. To achieve acceleration with the proposed method requires that the sequences are compressible. This is consistent with compressive computations in the discrete observation case, where for example computations for a non-repeating sequence of $N$ observations using $M = N$ different symbols cannot be accelerated. In either case, the highest possible acceleration is possible with constant sequences.

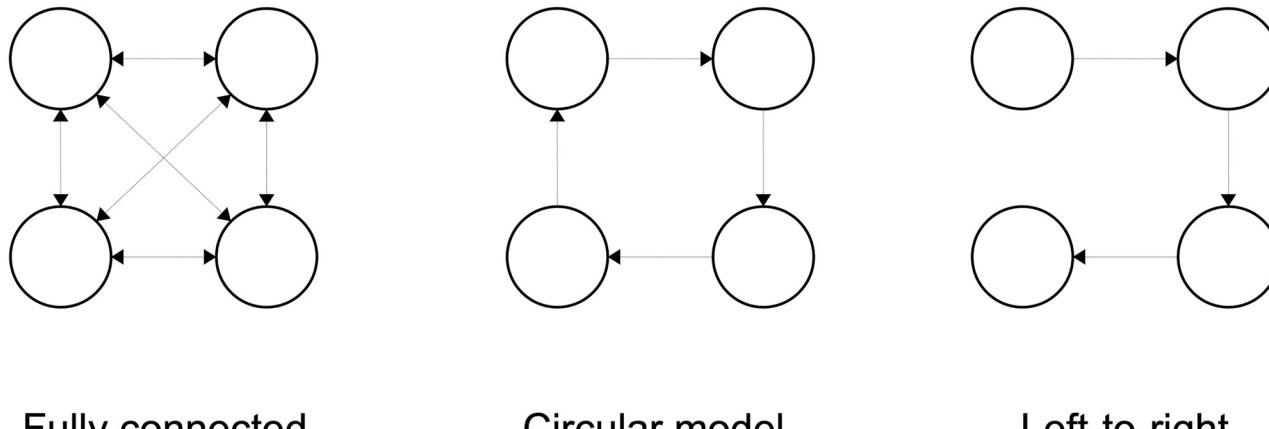

Fig 2. The HMM topologies used in the tests shown for *N* = 4 states. The simulations used $N \in \{2, 3, 5, 10, 20\}$ states. The presented conclusions will cover the fully-connected model due to other topologies yielding very similar results.

## Simulation study

We designed a large-scale simulation study to evaluate accuracy and speedups of the compressed approach. In total we created 50 different HMMs and simulated a total of 500 millions observations. The HMM topologies, reflecting popular choices in applications, belonged to one of three classes: fully-connected with transitions from state *i* to every state *j*; left-to-right, where a partial ordering on states exists and the possibility of a transition $i \rightarrow j$ implies $i \leq j$; circular, with transitions $i \rightarrow i + 1 \mod M$ or $i \rightarrow i$. See Fig 2 for details. For each topology, we choose a range of values for the number of states, $N \in \{2, 3, 5, 10, 20\}$. The transition probabilities were always chosen so that the average number of state transitions in a generated sequence would be 10*N*. Specifically, for fully-connected topologies the self-transition probabilties were defined to be $1 - (10N/T)$ and the non-self transition probabilties where chosen uniformly according to the topology, so that the total of outgoing probabilities sum up to 1. For left-to-right and circular topologies the self-transition and non-self-transitions were chosen similarly.

The identifyability of states depends on how different the univariate Gaussian state densities are. If the pair-wise differences of means are bounded from below by some $\epsilon > 0$ and all variances are 0, then the model simplifies to a Markov chain. On the other hand, the absolute values of the pair-wise differences only matter with respect to the variance. Consequently, the states' emission probabilities were taken as evenly distanced Gaussians starting from the standard Gaussian $Z(0, 1)$, with state separation $\eta_{S_1, S_2} = \frac{|\mu_{S_1} - \mu_{S_2}|}{3(\sigma_{S_1} + \sigma_{S_2})}$ varying from 0.1 to 1.0 in 0.1 increments, to obtain a range of emission densities corresponding to hard and easy scenarios for state identifyability.

For each HMM, a set of 100 tests was executed to average out the results: each test includes a data generation phase (with 100k observations), and a run of the evaluation, decoding and training algorithms. Evaluation and decoding was run once. As the training algorithm can yield quite different results based on the starting model estimation, in each test the training was repeated 10 times with different initial parameter estimates produced by the *K*-means algorithm. The Baum-Welch algorithm was set to terminate after 100 iterations or when the improvement on the evaluation probability (in the logarithmic space) was below $10^{-3}$.

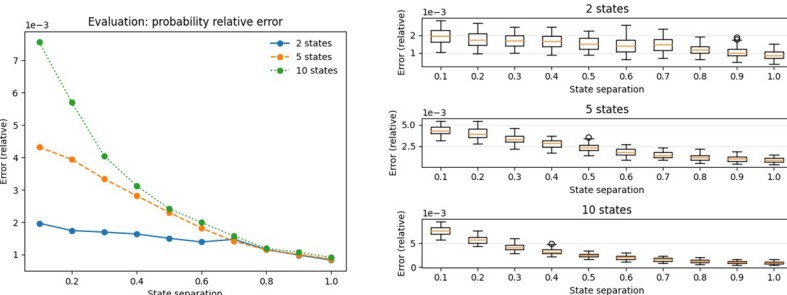

**Fig 3. We show the relative difference between the $P(O|\lambda)$ log-probabilities of the compressed and standard algorithms.** Since the errors are relative to a large magnitude negative log-probability, the actual error is extremely small. As an example, the relative error on the evaluation probability for a test involving a fully-connected model with 2 states is approximately 2e-3; the absolute error in the logarithmic space is around -6e2, which corresponds to a normal absolute error of approximately 1e-261.

## Observations

Here we will present observations from our simulation study. As we observed highly concordant results for all tested topologies we only present the results for fully-connected HMMs. As we obtained very favorable and highly consistent results for all simulations, we also present figures with additional diagnostics in the supplement.

For all three algorithms—evaluation, decoding and training—, the error compared to the uncompressed version is negligible regardless of the model topology. Fig 3 shows the error for the evaluation algorithm. Despite the overall negligible magnitude, the largest relative error displayed in Fig 3 is approximately 7e−3, which corresponds to an absolute error on the scale of $10^{-5000}$. There are some differences depending on model size and state separation, with larger models and lower state seperation displaying higher errors.

Having ascertained that the results of the compressed algorithms are virtually indistinguishable from the result of the standard algorithms, we investigated the attainable speedups. A general trend through all the tests is that the compressed algorithms are much faster then the standard ones, and that it particularly shows when the state separation is low. Evaluation and decoding results, see Figs 4 and 5 respectively, are again quite similar, as for both of them the compressed algorithm is several times faster than the standard one. In the evaluation task a 10-state model saw a speedup of 6–10 times, and a 5 state-model of 2-3 times compared to the uncompressed case. We observed no speedup for the 2-state model. A likely explanation is that the small size of the dynamic programming matrices and the model will allow very rapid branch-free in-cache computations, whereas the compressed version had a higher algorithmic overhead. The trends are very clear though in that the compression empirically reduces the running time from $O(N^2T)$ for the uncompressed version to something linear in both number of states and sequence length.

The most impressive speedups are obtained for the training algorithm, see Fig 5. There are three contributing factors to the speedup. First, as Baum-Welch training performs one forward and backward computation at $O(N^2T)$ per each individual reestimation step and then some additional computation on the order of $O(NT)$ we expect a speed-up as large as the one seen for the evaluation problem, i.e. of 2–10 depending on model size. Second, the benchmarking for the evaluation problem included the setup operations such as wavelet computation and population of the breakpoint array followed by *one* execution of the forward algorithm. In Baum-Welch training the setup costs are amortized over *multiple* forward and backward

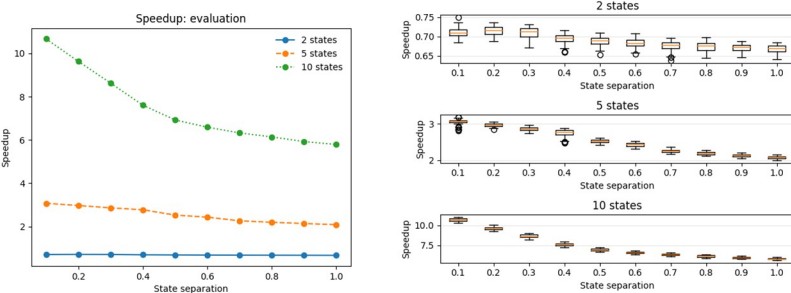

**Fig 4. Speedup on the evaluation problem using the compressed algorithm, including the input data processing time.**

computations, yielding an even larger acceleration over the uncompressed algorithm in this case. Third, we observe that the compression also has a large impact on the convergence, in particular for hard instances with low state separation. There, the compression greatly reduces the amount of reestimation steps needed, up to 10-fold for 10 states and state seperation 0.1.

## Discussion

In recent years, compression has become an effective computational building block to accelerate algorithms in response to ever-increasing data set sizes across a wide range of fields. The main idea is often to avoid re-computation for identical patterns in the data, as in the case for Hidden Markov Models (HMM) with discrete-valued observations whether for the frequentist classical three algorithms [9] or Gibbs sampling for Bayesian HMM [12].

For continous-valued observations, and for data resembling piece-wise constant functions plus Gaussian noise in the ideal case, our prior work [19, 20] on Gibbs sampling for Bayesian HMM has clearly shown how much can be gained from a compressive approach. The main insight there was that the compression—assuming that hidden states are not changed within contiguous blocks of the data with little variation—allows the Bayesian computation to focus on breakpoints, or state changes, between blocks instead of the variation within a block. This resulted not only in accelerated computations due to reducing the effort from something proportional to the number of blocks instead of the length of the observation sequence, but also a drastically accelerated convergence rate of the Gibbs sampling.

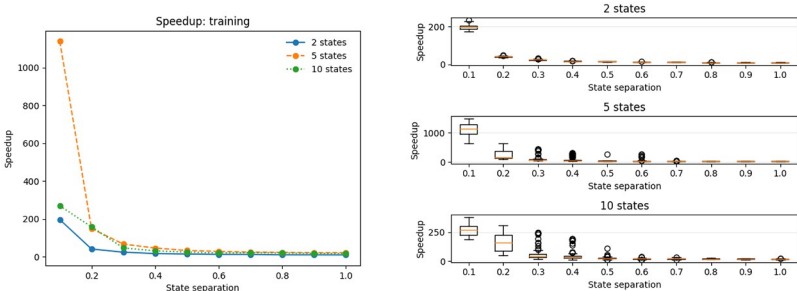

**Fig 5. Speedup on the training problem using the compressed algorithm, including the input data processing time.**

Here, we introduce the first compressive approach for Rabiner's three original problems [1] —likelihood computation, computing Viterbi paths, and Maximum-likelihood estimation with the Baum-Welch algorithm—for Hidden-Markov Models with continous emissions based on compression obtained with wavelets. We utilized the approach to compression developed in [19, 20] and extended it to the classical three algorithms. We used a wavelet transform to rapidly detect a superset of possible block boundaries in the observation sequence where state changes might occur, and adapted the computations of forward, backward and Viterbi variables accordingly by summing over blocks instead of individual observations.

In a large-scale simulation study, assuming a similar type of data, piece-wise constant functions plus homoscedastic Gaussian noise, we demonstrate that again we reach acceleration across all the fundamental dynamic programming algorithms and achieve practically relevant speed-up for training, likelihood, and Viterbi-path computations for models with 5 states and larger. In particular for training, we observe similar behaviour as for the convergence rate of the Gibbs sampler as in the Bayesian case [19]. There, enforcing that the hidden state variable for all observations in a block must be equal, prevented the sampler from exploring states where one or several of those variables where set to discordant values. In the Baum-Welch algorithm, the blocks seem to similarly prevent exploration of intermediate, sub-optimal states and thus to accelerate convergence. This is effect is clearly visible in the cases of low separability between state density means in contrast with high separability with total speedups—combining the contribution of accelerated computation of forward and backward variables and the contribution of accelerated convergence—of up to several hundred fold. The latter are almost trivial to resolve and uncompressed Baum-Welch converges also very rapidly, the impact of the compression on the convergence rate increasese as the separability decreases.

Note that the likelihood contribution from all observations in a block is de facto averaged and a decision about the best hidden state variable is made jointly for the block. If the data is indeed compressible, then the wavelet compression will indicate a superset of blocks and the adapted algorithm will find, as demonstrated empirically, the best hidden state variable per-block, which allows the training to converge rapidly.

As expected, the acceleration depends on compressibility of data and the number of states. Small models with only a few states benefit less than large models. We observed negligible influence of the model topology (not shown).

## Conclusion

Together with prior work we have demonstrated that compression can substantially accelerate frequentist and Bayesian HMM algorithms with both discrete and continuos observations as long as the data is amenable to compression. In the latter case, the method assumes that the data is piecewise constant with homoscedastic noise.

In future work it would be interesting to extend the approach to heteroscedastic noise, which requires static respectively dynamic adaptation of the noise threshold in the wavelet compression. For discrete observations, the counterpart to the current work is run-length encoding (RLE) and data compressible with RLE; note that compressed computations are exact in the discrete case. Mozes et al. [9] also proposed compression based on Lempel-Ziv parsing. Finding an analogue for continous observations might be a challenge, but possibly symbolic representation schemes developed for indexing time-series, e.g. [26], can provide a starting point.

## Acknowledgments

LB acknowledges support from the EU's Erasmus program for an exchange with Chalmers Technical University.

## Author Contributions

**Conceptualization:** John Wiedenhöft, Alexander Schliep.

**Formal analysis:** Luca Bello.

**Methodology:** Luca Bello, John Wiedenhöft, Alexander Schliep.

**Software:** Luca Bello.

**Supervision:** John Wiedenhöft, Alexander Schliep.

**Validation:** Luca Bello.

**Visualization:** Luca Bello.

**Writing – original draft:** Luca Bello, John Wiedenhöft, Alexander Schliep.

**Writing – review & editing:** Luca Bello, John Wiedenhöft, Alexander Schliep.

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
