## [Decision Letter · Decision Letter 0]

14 Feb 2023

PONE-D-22-22196Compressed computations using wavelets for hidden Markov models with continuous observationsPLOS ONE

Dear Dr. Schliep,

Thank you for submitting your manuscript to PLOS ONE. After careful consideration, we feel that it has merit but does not fully meet PLOS ONE’s publication criteria as it currently stands. Therefore, we invite you to submit a revised version of the manuscript that addresses the points raised during the review process.

We look forward to receiving your revised manuscript.

Kind regards,

Amita Nandal

Academic Editor

PLOS ONE

Journal Requirements:

Additional Editor Comments (if provided):

Authors need to adhere to the reviewer's comments

1. There should be a separate section in the manuscript highlighting the novelty of the proposed method.

2. The proposed method should be compared with other state-of-the-art techniques to highlight its effectiveness.

3.At the end of the introduction, it is difficult to understand your research motivation due to the lack of necessary summary and innovation of the method.

4.There is a lack of concatenation among the chapters of the article. It is suggested that the motivation between the chapters be supplemented.

5. Formula (3) lacks the introduction of variable definition and it is suggested to be supplemented.

6. It is suggested that the next research direction should be supplemented in the conclusion, and the reasonable prospect of this study should be carried out in order to inspire the follow-up scholars.

7.What are the meanings of these variables in formula (2)? Please add.

8. Why the method proposed by the author has achieved the best effect, please make a specific analysis.

Reviewers' comments:

Reviewer's Responses to Questions

**Comments to the Author**

1. Is the manuscript technically sound, and do the data support the conclusions?

Reviewer #1: Yes

Reviewer #2: Yes

2. Has the statistical analysis been performed appropriately and rigorously? 

Reviewer #1: Yes

Reviewer #2: Yes

3. Have the authors made all data underlying the findings in their manuscript fully available?

Reviewer #1: Yes

Reviewer #2: Yes

4. Is the manuscript presented in an intelligible fashion and written in standard English?

Reviewer #1: Yes

Reviewer #2: Yes

5. Review Comments to the Author

Reviewer #1: The paper is well written and has some novelty. Following are the comments-

1. There should be a separate section in the manuscript highlighting the novelty of the proposed method.

2. The proposed method should be compared with other state-of-the-art techniques to highlight its effectiveness.

Reviewer #2: 1.At the end of the introduction, it is difficult to understand your research motivation due to the lack of necessary summary and innovation of the method.

2.There is a lack of concatenation among the chapters of the article. It is suggested that the motivation between the chapters be supplemented.

3. Formula (3) lacks the introduction of variable definition and it is suggested to be supplemented.

4. It is suggested that the next research direction should be supplemented in the conclusion, and the reasonable prospect of this study should be carried out in order to inspire the follow-up scholars.

5.What are the meanings of these variables in formula (2)? Please add.

6. Why the method proposed by the author has achieved the best effect, please make a specific analysis.

6. PLOS authors have the option to publish the peer review history of their article (what does this mean?). If published, this will include your full peer review and any attached files.

Reviewer #1: No

Reviewer #2: No

---

## [Author Response · Author response to Decision Letter 0]

1 Mar 2023

Response to Reviewers

PONE-D-22-22196 Compressed computations using wavelets for hidden Markov models with continuous observations

In the following I will address the reviewer’s comments designated as mandatory by the academic editor in detail. It seems that we failed to communicate sufficiently clearly that our method is the very first and only method to allow compressive computations for frequentist HMM algorithms with continuous emissions (cf. below). We added exposition to the manuscript to clarify this point.

For the authors, Alexander Schliep

1. There should be a separate section in the manuscript highlighting the novelty of the proposed method.

The novelty of the method has previously been highlighted in the final paragraph of the introduction. We added a section header to indicate it.

2. The proposed method should be compared with other state-of-the-art techniques to highlight its effectiveness. 

Ours is the first method to perform compressed computations of Likelihood, Viterbi paths, and Maximum-likelihood estimation with the Baum-Welch algorithm for HMMs with continuous emissions. Prior approaches, including ours for Bayesian HMMs with discrete and continuous emissions, tackle different problems or different types of emissions. As such, our method is the only method for this problem, making an evaluation benchmark impossible as no competing methods exist.

We added some text to abstract, introduction and conclusion to make this more obvious.

3.At the end of the introduction, it is difficult to understand your research motivation due to the lack of necessary summary and innovation of the method. 

To summarize, the research motivation is as follows. Compressed computations for Hidden Markov Models have been used with great success for three out of the four cases resulting from two types of emissions (discrete vs. continuous) and two types of computations (frequentist—Viterbi, likelihood and Baum-Welch—vs. Bayesian MCMC) by us and one other group. The submission focuses on the fourth and last open case (frequentist and continuous). This has been motivated in the abstract “Here we extend the compressive computation approach for the first time to the classical frequentist HMM algorithms on continuous-valued observations” . In the introduction we give a complete summary of the subfield “Mozes et al. ... for discrete observations achieve considerable speed-ups”, “Also for discrete observations, Mahmud et al. substantially accelerate Forward-Backward Gibbs

(FBG) sampling ...” and “ substantial improvement [18] in the running times of the FBG sampler for continuous-valued observation”, to “In the following, we introduce compressive computations based on wavelets for Rabiner’s three original problems [1]: likelihood computation, computing Viterbi paths, and Maximum-likelihood estimation with the Baum-Welch algorithm.” We also added corresponding language to the discussion.

The last paragraph of the introduction is a succinct, complete and exhaustive summary of what we accomplish in the manuscript, now highlighted by a separate section header.

4.There is a lack of concatenation among the chapters of the article. It is suggested that the motivation between the chapters be supplemented. 

We added some additional exposition to the manuscript.

5. Formula (3) lacks the introduction of variable definition and it is suggested to be supplemented. 

It is not clear to us which variable(s) the reviewer might be referring to. Every variable used in equations (3a)–(3b) has been defined and explained in lines 86–97, resp. lines 72–84 (conditional probability), and line 102 (the likelihood function). We additionally made the definition of y_s^t explicit by repeating and adapting the definition of q_s^t.

6. It is suggested that the next research direction should be supplemented in the conclusion, and the reasonable prospect of this study should be carried out in order to inspire the follow-up scholars.

As a matter of fact we already do discuss potential follow-up research in the conclusion (note, we follow PLOS ONE suggested article structure), in particular suggesting an extension to heteroscedastic noise and a potential use of the symbolic indexing by Keogh et al..

7.What are the meanings of these variables in formula (2)? Please add.

It is not clear to us which variable(s) the reviewer might be referring to. Every variable used in equation (2) has been defined and explained in lines 86–97, resp. lines 72–84 (conditional probability), and line 102 (the likelihood function). We additionally made the definition of y_s^t explicit by repeating and adapting the definition of q_s^t.

8. Why the method proposed by the author has achieved the best effect, please make a specific analysis.

We don’t make the claim that our method has the “best effect”, as we do not perform a benchmark evaluation against other methods. The reason why we don’t perform a benchmark evaluation is that ours is the first method to perform compressed computations of likelihood, Viterbi paths, and Maximum-likelihood estimation with the Baum-Welch algorithm for HMMs with continuous emissions. Prior approaches, including ours for Bayesian HMMs with discrete and continuous emissions, tackle different problems or different types of emissions. As such, our method is the only and thus the state-of-the-art method for this problem, making an evaluation benchmark impossible as no competing method exists.

What we did benchmark is a comparison of uncompressed (prior art) and compressed (our contribution) versions; to reiterate there are no other compression method for the continuous case and the particular HMM algorithm we consider.

---

## [Decision Letter · Decision Letter 1]

9 May 2023

Compressed computations using wavelets for hidden Markov models with continuous observations

PONE-D-22-22196R1

Dear Dr. Schliep,

We’re pleased to inform you that your manuscript has been judged scientifically suitable for publication and will be formally accepted for publication once it meets all outstanding technical requirements.

Kind regards,

Amita Nandal

Academic Editor

PLOS ONE

Additional Editor Comments (optional):

The manuscript can be accepted for publication.

Reviewers' comments:

Reviewer's Responses to Questions

**Comments to the Author**

1. If the authors have adequately addressed your comments raised in a previous round of review and you feel that this manuscript is now acceptable for publication, you may indicate that here to bypass the “Comments to the Author” section, enter your conflict of interest statement in the “Confidential to Editor” section, and submit your "Accept" recommendation.

Reviewer #2: All comments have been addressed

Reviewer #3: All comments have been addressed

2. Is the manuscript technically sound, and do the data support the conclusions?

Reviewer #2: Yes

Reviewer #3: Yes

3. Has the statistical analysis been performed appropriately and rigorously? 

Reviewer #2: Yes

Reviewer #3: Yes

4. Have the authors made all data underlying the findings in their manuscript fully available?

Reviewer #2: Yes

Reviewer #3: Yes

5. Is the manuscript presented in an intelligible fashion and written in standard English?

Reviewer #2: Yes

Reviewer #3: Yes

6. Review Comments to the Author

Reviewer #2: The authors of the article have made extensive edits. It's finally ready for journal publishing, in my opinion.

Reviewer #3: The authors have revised the paper significantly. I think it can be now accepted for publication in the journal.

7. PLOS authors have the option to publish the peer review history of their article (what does this mean?). If published, this will include your full peer review and any attached files.

Reviewer #2: No

Reviewer #3: No

---

## [Editor Report · Acceptance letter]

29 May 2023

PONE-D-22-22196R1 

Compressed computations using wavelets for hidden Markov models with continuous observations  

Dear Dr. Schliep:

I'm pleased to inform you that your manuscript has been deemed suitable for publication in PLOS ONE. Congratulations! Your manuscript is now with our production department. 

Kind regards, 

on behalf of

Dr. Amita Nandal 

Academic Editor

PLOS ONE